# Multi-Modal Compositional Analysis of Layered Paint Chips of Automobiles by the Combined Application of ATR-FTIR Imaging, Raman Microspectrometry, and SEM/EDX

**DOI:** 10.3390/molecules24071381

**Published:** 2019-04-08

**Authors:** Md Abdul Malek, Takashi Nakazawa, Hyun-Woo Kang, Kouichi Tsuji, Chul-Un Ro

**Affiliations:** 1Department of Chemistry, Inha University, Incheon 22212, Korea; abdulmalek1654@gmail.com (M.A.M.); chunsangar@naver.com (H.-W.K.); 2Graduate School of Engineering, Osaka City University, Osaka 558-8585, Japan; takashi.nakazawa@cem.com (T.N.); tsuji@a-chem.eng.osaka-cu.ac.jp (K.T.)

**Keywords:** layered car paint chips, SEM/EDX, ATR-FTIR imaging, Raman microspectrometry

## Abstract

For the forensic analysis of multi-layered paint chips of hit-and-run cars, detailed compositional analysis, including minor/trace chemical components in the multi-layered paint chips, is crucial for the potential credentials of the run-away car as the number of layers, painting process, and used paints are quite specific to the types of cars, color of cars, and their surface protection depending on the car manufacturer and the year of manufacture, and yet overall characteristics of some paints used by car manufacturers might be quite similar. In the present study, attenuated total reflectance-Fourier transform infrared (ATR-FTIR) imaging, Raman microspectrometry (RMS), and scanning electron microscopy/energy-dispersive X-ray spectrometric (SEM/EDX) techniques were performed in combination for the detailed characterization of three car paint chip samples, which provided complementary and comprehensive information on the multi-layered paint chips. That is, optical microscopy, SEM, and ATR-FTIR imaging techniques provided information on the number of layers, physical heterogeneity of the layers, and layer thicknesses; EDX on the elemental chemical profiles and compositions; ATR-FTIR imaging on the molecular species of polymer resins, such as alkyd, alkyd-melamine, acrylic, epoxy, and butadiene resins, and some inorganics; and RMS on the molecular species of inorganic pigments (TiO_2_, ZnO, Fe_3_O_4_), mineral fillers (kaolinite, talc, pyrophyllite), and inorganic fillers (BaSO_4_, Al_2_(SO_4_)_3_, Zn_3_(PO_4_)_2_, CaCO_3_). This study demonstrates that the new multi-modal approach has powerful potential to elucidate chemical and physical characteristics of multi-layered car paint chips, which could be useful for determining the potential credentials of run-away cars.

## 1. Introduction

When a car traffic accident occurs, pieces of paint chips from the surface coating of the cars can be scraped out and remain on the spot, even when the car runs away from the site. In this case, an analysis of the paint chips is important for forensic investigations [1,2]. Car paint chips generally have a layered structure, resulting from a painting process. As the painting process and used paints are specific to the types of cars, color of cars, and surface protection, depending on the car manufacturer and the year of manufacture, and yet the overall characteristics of some paints used by car manufacturers might be similar, detailed compositional analysis, including minor/trace chemical components in multi-layered paint chips, can be crucial for the potential credentials of the run-away car.

For chemical analysis of coating and paint samples of automobiles, various techniques, such as laser ablation inductively coupled plasma mass spectrometry (LA-ICP-MS) [3], laser-induced breakdown spectroscopy (LIBS) [4], pyrolysis gas chromatography-mass spectrometry (GC-MS) [5], and time-of-flight secondary ion mass spectrometry (TOF-SIMS) [6], have been employed as destructive analytical tools, while micro X-ray fluorescence (micro-XRF) [7,8], scanning electron microscopy/energy-dispersive X-ray spectrometry (SEM/EDX) [9], Raman microspectrometry (RMS) [10], Fourier transform infrared spectroscopy (FTIR) [11], synchrotron FTIR [12], and attenuated total reflectance FTIR (ATR-FTIR) in conjunction with multivariate chemometrics [13] have been reported to be more promising for forensic purposes due to their non-destructive or semi-destructive nature. On the other hand, the application of a single analytical technique is not sufficient to extract the comprehensive information because commercial paints are complex mixtures [14] of organics (polymer dyes, binders, additives, etc.), inorganics (pigments, fillers, elemental C or Al flakes, etc.), and sometimes minerals. Thus, multiple techniques would be more useful for the analysis of car paint chips. Analytical techniques, such as FTIR spectroscopy together with SEM/EDX and/or micro-XRF [15,16], RMS together with SEM/EDX and/or micro-XRF [17], FTIR together with RMS and micro-XRF [18,19], and ATR-FTIR (point mode) together with RMS and SEM/EDX [20,21], have become promising because of their rapid, quantitative ability and confirmatory result. FTIR spectroscopy is a popular technique and most FTIR studies have been based on transmission FTIR spectroscopy measurements because of the widely available transmission FTIR library spectra [15,16,18,19]. Despite this, transmission FTIR measurements demand the sophisticated preparation of ultrathin samples. On the other hand, ATR-FTIR measurements in point mode demand the physical separation of individual layers (using a scalpel), which is difficult to do without mixing up the micro-sized layers, or multiple measurements are needed for successive layers, where focusing of the IR beam on each layer through an internal reflection element (IRE) crystal might not be easy or reliable for micro-sized thin layers. In contrast, ATR-FTIR imaging measurements of polished cross-sections of embedded paint samples can be more practical as they can provide information on the spatially distributed chemicals within samples and be applied conveniently to characterize the micro-sized layers of car paint chips [22,23,24,25]. However, to the best of the authors’ knowledge, the application of ATR-FTIR imaging together with SEM/EDX or RMS has never been carried out for multi-layer car paint analysis. 

SEM/EDX can provide information on the physical structures and elemental compositions of micrometer-sized samples with submicron lateral resolution, and yet it has limited capabilities for performing molecular speciation of particles. Vibrational spectroscopic techniques, such as RMS and ATR-FTIR, are powerful for functional group analysis and molecular speciation of organic and inorganic chemical compounds, including hydrated species, under ambient conditions. Although RMS and ATR-FTIR are similar in that they belong to vibrational spectroscopic techniques, their vibrational signals are generated from different fundamentals; i.e., RMS provides information on molecular vibrations based on the difference in wavelength between the incident and scattered visible radiation (Raman scattering), whereas ATR-FTIR is based on the attenuation of the evanescent wave generated by the total reflected mid-IR radiation on the IRE crystal. According to selection rules, for IR spectroscopy, it is necessary for the molecule to have a permanent electric dipole, and for Raman spectroscopy, it is the polarizability of the molecule which is important. Therefore, the differences in their spectra owing to their different signal generation mechanisms (i.e., scattering vs. absorption of energy) and different selection rules would make two fingerprint techniques rather complementary. RMS and ATR-FTIR imaging provide spectra with a typical spectral range between 50 and 4000 cm^−1^ and 680 to 4000 cm^−1^, respectively, making RMS efficient to identify metal oxides showing their peaks at the far-IR region. Further, due to the incident radiation, RMS has better lateral resolution than ATR-FTIR imaging has, so that RMS is more powerful for the investigation of heterogeneity of micrometer-sized samples. The mostly sharp Raman peaks are useful for unambiguous molecular speciation. On the other hand, the laser beam employed in RMS can induce damage to the samples and the interference by the fluorescence often encountered in RMS needs to be minimized, which are not problems in ATR-FTIR measurements. ATR-FTIR imaging provides the ATR-FTIR spectrum at each pixel in the image field for the samples, whereas RMS mapping to obtain the spatial, chemical heterogeneity for the samples takes a much longer time as RMS images are acquired by the point-by-point scanning mode [26,27,28,29,30,31,32,33]. 

Herein, ATR-FTIR imaging, RMS, and SEM/EDX were used in combination for compositional analysis of three samples of multi-layered car paint chips. The unambiguous molecular speciation of the chemical components within the layers as well as a determination of the physical layered structures were possible using this multi-modal approach. ATR-FTIR imaging and Raman spectral data can provide information on the molecular species of organic polymer resins and inorganic compounds, and on their spatial distributions on a micrometer scale. Although information on the elemental composition from SEM/EDX analysis is insufficient for molecular speciation, the detection of chemical elements in the layers is consistent with and supportive of the ATR-FTIR imaging and Raman data on the polymer resins, inorganics, and pigments. This multi-modal approach has great potential to investigate the chemical and physical characteristics of car paint chips in detail, which would be powerful for elucidating the potential credentials of cars involved in hit-and-run accidents.

## 2. Materials and Methods

### 2.1. Samples

Three car paint chips (samples A to C as shown in Figure 1a–c, respectively) were provided by the Forensic Science Laboratory of Hyogo Prefectural Police Headquarters (Hyogo, Japan). The numbers of layers of samples A to C were confirmed to be 9, 5, and 4, respectively, by optical, SEM, and ATR-FTIR imaging (Figure 2). Reagent grade chemicals used as standard reference materials for ATR-FTIR imaging and RMS measurements, such as rutile (TiO_2_), anatase (TiO_2_), BaSO_4_, ZnO, Zn_3_(PO_4_)_2_, Fe_3_O_4_, and CaCO_3_, were purchased from Sigma Aldrich (Darmstadt, Germany). Standard minerals, such as kaolinite (KGa-1b), were obtained from the International Clay Mineral Society (ICMS), while talc and pyrophyllite were collected from the Korea Institute of Geoscience and Mineral Resources (KIGAM).

### 2.2. Sample Preparation for ATR-FTIR Imaging and RMS Measurements

As shown in Figure 1d–f, which present optical images of the polished molds of samples A to C, respectively, first, the three paint chip samples, approximately 2.5 × 1.5 mm^2^ in size, were held vertically using double ringed heart-shaped stainless-steel clips. The clips holding the paint chips were placed into plastic petri dishes, and liquid epoxy resin (bisphenol-A-epichlorohydrin) mixed with a hardening agent (triethylenetetramine) was poured until the clips were embedded in the resin. When the resin solidified, the plastic cases were broken, and solid cylinder-shaped molds were recovered. During molding, the cross-sectional surfaces of the paint chips were covered with resin. The surfaces of the molds were polished using a TwinPrep3^TM^ polisher (Allied High Tech Products, Inc., Rancho Dominguez, CA, USA) to uncover the layered cross-sectional surface of the paint chips. Multiple-step polishing was performed to obtain smooth cross-sectional surfaces of the paint chips without destroying their layered structures. First, sand papers with 200, 800, 1200, 2000, and 2400 mesh numbers were used sequentially for gradual fine polishing, where the surface of the molds was maintained manually in gentle contact against rotating sand paper attached to the rotating plate of the polisher. Second, a water-based polycrystalline diamond suspension was spread over a polishing cloth attached to the rotating plate of the polisher against which the sample was maintained in gentle contact. The diamond suspensions with different particle sizes of 6.0, 3.0, 1.0, and 0.5 µm in diameter were used successively to smooth the surface. This rigorous sample preparation was performed to maintain exact multi-layer sample structures of a micrometer size and thus to ensure the reliable investigation of the samples using the multi-modal approach. The overall thicknesses of the cross-sections of samples A to C were ~260, 145, and 95 μm, respectively, based on the secondary electron images (SEIs) from SEM (Figure 2d–f). The cross sections of the molded and polished paint surfaces were investigated sequentially by ATR-FTIR imaging, RMS, and SEM/EDX, so that the conductive C-coatings of the samples, which were applied prior to the SEM/EDX measurement to reduce charge accumulation, would not interfere with the ATR-FTIR imaging and RMS measurements.

### 2.3. ATR-FTIR Imaging Measurements

ATR-FTIR imaging measurements of cross-sections of the three molded layered samples were performed using a Perkin-Elmer Spectrum 100 FTIR spectrometer interfaced to a Spectrum Spotlight 400 FTIR microscope (Waltham, MA, USA). For ATR-FTIR imaging, an ATR accessory employing a germanium hemispherical IRE crystal, 600 μm in diameter, was used. As shown by the red dots in Figure 1d–f, suitable positions (Figure 2a–c) on the cross-sections were located using a visible light optical microscope equipped with a light-emitting diode and a charge-coupled device (CCD) camera, and the ATR-FTIR imaging measurements were performed. A description of the sample measurement procedure can be found elsewhere [26,27]. A spatial resolution of 3.1 μm at 1726 cm^−1^ (λ = 5.79 μm) is achievable [28]. A 16 × 1 pixel mercury cadmium telluride (MCT) array detector was used to obtain the FTIR images with a pixel size of 1.56 μm. For each pixel, an ATR-FTIR imaging spectrum, ranging from 680 to 4000 cm^−1^ with a spectral resolution of 4 cm^−1^, was obtained from eight interferograms that were co-added and Fourier-transformed. The ATR-FTIR images (Figure 2g–i) were obtained by principal component analysis (PCA) after the first differentiation of the original ATR-FTIR imaging spectra at all pixels in the raw images. The representative ATR-FTIR spectrum for each layer was extracted at the center of the layer if the layer was chemically homogeneous. Otherwise, the spectra obtained throughout the layer were analyzed to investigate its heterogeneity. The molecular speciation based on the ATR-FTIR spectra was performed primarily utilizing the Perkin-Elmer FTIR spectral libraries for spectral library searching.

### 2.4. RMS Measurements

The molded and polished samples were placed on the microscope stage so that the layered cross-sections remained perpendicular to the laser beam of a confocal Raman microspectrometer (XploRA, Horiba Jobin Yvon, Horiba-Scientific, Lille, France) equipped with a 100×, 0.9 numerical aperture objective (Olympus, Tokyo, Japan). The Raman measurements were performed using a diode laser with a wavelength of 785 nm, and the Raman signals were detected using a CCD detector. The laser power delivered to the sample was approximately 12 mW, which could be attenuated by a set of neutral density filters with an optical density ranging from 100% to 0.1%. The spot size of the laser at the sample was estimated to be ~1 µm^2^. The acquisition of the spectra and images, and the peak assignments were achieved using Labspec6 software. For each analysis, data acquisition was performed over the spectral range of 50 to 2000 cm^−1^ with a spectral resolution of 4 cm^−1^. The computer-controlled XYZ Raman microscope stage allowed a line scan with a 1 µm step size across the cross-sections of the samples. More detailed discussions of the measurement conditions can be found elsewhere [29,30,31].

### 2.5. SEM/EDX Measurements

SEM/EDX was used to determine the elemental composition information on the molded, polished, and C-coated cross-sections (Figure 2d–f). The measurements were carried out using a Jeol JSM-6390 SEM equipped with an Oxford Link SATW ultrathin window EDX detector with a resolution of 133 eV for the Mn-Kα X-rays. The X-ray spectra were recorded under the control of Oxford INCA Energy software. Using a 20 keV accelerating voltage and a 0.5 nA beam current, the line scan analysis mode was carried out using a 1 µm step for EDX data acquisition. More detailed discussions of the measurement conditions can be found elsewhere [29,30,31].

## 3. Results and Discussion

Owing to the complex nature of the individual components present in real coating and paint samples [14], their unambiguous characterization is challenging. For example, epoxy resins can exist in a variety of chemical structures in terms of the presence or absence of aromatic rings, the length and nature of the aliphatic backbone chain, the positions of group substitutions, the degree of polymerization, etc. Detailed characterization of the individual layers cannot be comprehensibly achieved using a single analytical technique because the three car paint chip samples examined in this study also possess complicated characteristics in terms of the number of layers, layer thicknesses, and compositions. 

### 3.1. Characterization of Sample A

Visible light optical, secondary electron (SE), and ATR-FTIR images of the molded and polished cross-section of sample A (Figure 2a,d, and g, respectively) revealed nine layers with different layer thicknesses. Figure 3, Figure 4 and Figure 5 present the representative ATR-FTIR, Raman, and X-ray spectra of these layers, respectively. Based on those spectral data, five different types of resins, i.e., alkyd-based enamel resin along with polyalkoxylated phenol (PAP) derivatives (layers #1 and #4), alkyd-melamine resins (layers #2 and #5), acrylic-melamine resins (layers #3, #6, and #7), ester epoxy resin (layer #8), and non-ester epoxy resins (layer #9), were identified as the major organic species in sample A. Kaolinite (layers #3 and #6), talc and Al_2_(SO_4_)_3_ (layer #7), pyrophyllite (layer #8), and Al metal flakes (layers #2 and #5) were detected as additives or fillers. Based on the SEM/EDX and RMS data, rutile (layers #2, #3, #5, #6, and #7), Fe_3_O_4_ (layers #8 and #9), and probably SnO_2_ (layer #8) and ZnO (layer #9) were identified as the pigments. Detailed characteristics of each layer are discussed below.

#### 3.1.1. Characterization of Layers #1, #2, #4, and #5 (Alkyd and Alkyd-Melamine Resins)

Appendix A presents representative ATR-FTIR spectra of layers #1, #2, #4, and #5 and two library spectra with the best match, clearly showing the presence of alkyd-based enamel resins [33,34,35] as the major organic species in these layers. The Raman spectral data also support the presence of alkyd resins in these four layers [36]. Although the four layers are the same in terms of their major alkyd resin contents, layers #2 and #5 contain small quantities of melamine, whereas layers #1 and #4 do not. Comparatively strong doublet ATR-FTIR bands at ~1557 and ~1542 cm^−1^ and bands at 1296 and 814 cm^−1^ of layers #2 and #5 (ascribed by a solid heart sign in inset (a) of Appendix A) were assigned to the –C=N, asymmetric –C–N, symmetric –C–N, and –NH–CH_2_– stretching bands of melamine, respectively. In addition, the Raman peak at 974 cm^−1^ for layers #2 and #5 (shown by a double asterisk in Figure 4) is characteristic of the ring breathing modes, i.e., δ(CNC) and δ(NCN), of the substituted triazine ring of melamine [37,38]. In addition, the small N X-ray peaks observed for layers #2 and #5 (shown in an inset of Figure 5) support the presence of melamine. The addition of melamine in modern paints was reported to improve the coating properties [34,35,39] by cross-linking with alkyd [34,35], epoxy [40], acrylic [41], or polyester [42] resins. On the other hand, layers #1 and #4 appear to contain PAP surfactant as a solubilizer. As shown in insets (a) and (b) of Appendix A, the ATR-FTIR spectra of layers #1 and #4 reveal the characteristic peaks for PAP at 3060 and 3026 cm^−1^ (aromatic =C–H stretches), 1386 cm^−1^ (–CH_3_ bending (umbrella mode) in the presence of the alkoxy group), 1127 cm^−1^ (secondary > C–O stretching), 1050 cm^−1^ (primary C–C–O stretching), and 840 cm^−1^ (aromatic =C–H out-of-plane bending) [21,43,44]. The ATR-FTIR peaks for the aromatic C=C–C stretching or ring modes were observed at 1601, 1582, 1506, and 1493 cm^−1^ for all four layers, probably due to the presence of a phthalic group in the alkyd resins [21,45,46], where a peak pair at 1601 and 1493 cm^−1^ were assigned to the mono or ortho-substituted ring, and a peak at 1506 cm^−1^ was attributed to a para-substituted ring [47]. The Raman spectra of layers #1 and #4 showed additional peaks at ~798 and 1246 cm^−1^ (indicated by the asterisk in Figure 4), which are probably due to PAP in these two layers.

As shown in inset (a) of Figure 5, the X-ray spectra of layers #2 and #5 indicate the presence of Al and Ti, of which the X-ray intensities vary at different locations of the layers, suggesting that the Al and Ti moieties are embedded as dispersed entities. As shown in Appendix A, the Raman spectrum of standard rutile with sharp peaks at 610 and 446 cm^−1^ along with a broader peak centered at 234 cm^−1^ matched well with those of layers #2 and #5, which is different from that of standard anatase (another TiO_2_ polymorph) with peaks at 639, 515, and 395 cm^−1^ along with a very strong peak at 141 cm^−1^ [48]. The Raman peaks at 140 and 513 cm^−1^ in the standard rutile spectrum might be due to the presence of a small amount of anatase as an impurity. A dramatic increase in the Al X-ray intensity (see X-ray spectrum notated as #2(iii) in inset (a) of Figure 5) acquired at a white strip on SEI (inset (b) of Figure 5) suggests that Al exists as Al metal flakes. 

Alkyd-based enamel resins are the major component in layers #1, #2, #4, and #5, with small quantities of PAP solubilizer for layers #1 and #4 and with minor amounts of melamine for layers #2 and #5. As shown in Figure 3, the ATR-FTIR intensity of the C=O peak at ~1725 cm^−1^ was weaker for layer #1 than for layer #4 and weaker for layer #2 than for layer #5. This is probably because the outer layers (layers #1 and #2) had experienced more weathering and ageing than layers #4 and #5, respectively.

#### 3.1.2. Characterization of Layers #3, #6, and #7 (Acrylic-Melamine Resins)

As shown in Appendix A, a comparison of the ATR-FTIR spectra of layers #3, #6, and #7 as well as two library spectra with the best match indicated that the major organic species in these layers are acrylic resins. A strong peak at 1231 cm^−1^ and a doublet at 1181 and 1165 cm^−1^ are characteristic of acrylics in the fingerprint region [49,50]. In addition, the ATR-FTIR peaks (marked by a solid heart shape in Appendix A) at 1554, 1540, 1507, 1296, and 815 cm^−1^ indicate the presence of melamine in these three layers. In addition, the Raman spectra of these layers (Figure 4 and Appendix A) are consistent with the literature spectra of acrylic resins [51].

As shown in Figure 5, the Ti X-ray peaks were observed for layers #3, #6, and #7; Al and Si for layers #3 and #6; and Mg and Si and/or Al and S for layer #7. The Ti moiety is rutile, as confirmed by RMS (Appendix A) and ATR-FTIR (sloping, continuously decreasing peaks, marked by the asterisks in Appendix A). The Al and Si moiety in layers #3 and #6 is kaolinite, which is supported by their characteristic ATR-FTIR peaks for kaolinite at 937, 912, 1006, 1029, 3617, 3647, 3668, and 3668 cm^−1^ [52,53]. The X-ray peaks for Ti, Al, and Si were observed throughout layers #3 and #6, where the Ti peak intensity increased as the Al and Si intensity decreased and vice versa (see Appendix A), indicating that the rutile pigment and the kaolinite mineral were dispersed homogeneously, but either the pigment or the mineral was detected dominantly at different locations of the layers because they exist as separate, sizable particles. 

As shown in Figure 5 and Appendix A, the X-ray peak intensities of Mg and Si are rather random at different locations of layer #7, whereas the Al and S peak intensities are somewhat consistent, suggesting that Mg and Si, and Al and S are from different chemical moieties. The ATR-FTIR spectra of #7(a) and #7(b) in Appendix A indicate that the Mg and Si moiety is talc (the characteristic peak is observed at 1014 cm^−1^; notated as a solid triangle on the ATR-FTIR spectrum of #7(a)) and the Al and S are probably Al_2_(SO_4_)_3_ due to the SO_4_^2-^ peak at 1072 cm^−1^ (notated as blank triangles on the ATR-FTIR spectra of #7(a) and #7(b)). Because layers #3, #6, and #7 contain a significant amount of rutile with high Raman sensitivity, the Raman peaks of rutile are dominant in the RMS spectra of the layers (Figure 4). Therefore, RMS was not useful for detecting minor quantities of kaolinite, talc, and Al_2_(SO_4_)_3_.

#### 3.1.3. Characterization of Layers #8 and #9 (Epoxy Resins)

As shown in Appendix A, a comparison of the ATR-FTIR spectra of layers #8 and #9 and the epoxy resin mold used to prepare the cross-section samples in this study and two library spectra with the best match indicated that the major organic species in layers #8 and #9 are epoxy resins. The X-ray spectra obtained at various locations of layer #8 (Appendix A) revealed Al, Si, and Ti along with a small amount of Sn and trace amounts of Fe and Zn, which were observed consistently throughout the layer. Al and Si were from pyrophyllite as the ATR-FTIR peaks over the 1150 to 900 cm^−1^ range resemble those of standard pyrophyllite (Appendix A). In addition, a sloping decreasing peak with a peak minimum at 684 cm^−1^ ascribed by the asterisk indicates TiO_2_. Sn can be from SnO_2_ and the trace amount of Zn appears to be from a mineral impurity. On the other hand, layers #8 and #9 would be burnt out easily by absorbing Raman laser light. Hence, the Raman spectra obtained with a minimum laser power could only provide Raman peaks partially matching with the Fe_3_O_4_ standard (Appendix A) [54].

As shown in Appendix A, the X-ray spectra obtained at various locations of layer #9 indicate that the layer is composed of two sub-layers, where Al, Si, Ti, Zn, and P were detected in the inner sub-layer (near the boundary of layer #8 at a thickness of 251 to 255 μm), whereas only Zn and Fe were detected in the outer sub-layer (256~260 µm). Appendix A plots the ATR-FTIR spectra acquired at successive pixels on a line (*x*-axis) across the boundary between layers #8 and #9 after a baseline correction. Spectrum #8(i), which is representative of layer #8, shows a relatively stronger peak at 1039 cm^−1^, which resembles the main peak of the Si-O stretching of pyrophyllite. As the pixels move from layer #8 to layer #9 and cross the boundary, the peak positions of the Si–O stretching vibration shifted towards a lower wavenumber from ~1040 to 1026 cm^−1^ and the relative intensity of the shoulder peak at ~1010 cm^−1^ became gradually stronger, which might be due to the presence of kaolinite in layer #9 instead of pyrophyllite in layer #8. The ATR-FTIR peaks at ~954 cm^−1^ shifted to 935 cm^−1^ (spectra #9(i)–#9(v)) and gradually became stronger, probably due to the presence of PO_4_^3-^ group, corresponding to the observed P X-ray peak at this region. Therefore, the combination of ATR-FTIR, X-ray, and Raman data indicates the presence of kaolinite, Zn_3_(PO_4_)_2_, and TiO_2_ in the inner sub-layer of layer #9 and Fe_3_O_4_ and ZnO in the outer sub-layer. 

### 3.2. Characterization of Sample B

The visible light optical, SE, and ATR-FTIR images of the molded and polished cross-section of sample B (Figure 2b,e, and h, respectively) revealed fvie layers with different thicknesses. Figure 6, Figure 7, and Figure 8a present the representative ATR-FTIR, Raman, and X-ray spectra of the layers, respectively. Based on the spectral data, four types of resin, i.e., ester-epoxy resin (layer #1), alkyd-melamine resin of type-I (layers #2 and #3), alkyd-melamine resin of type-II (layer #4), and acrylic resin (layer #5), were identified as the major organic species in sample B. Kaolinite (layer #1) and BaSO_4_ (layers #2 and #3) were detected as fillers, whereas rutile, FeS_x_, and SnO_2_ (layer #1) and probably CuCl_2_ (layer #3) were detected as pigments. A detailed characterization of each layer is discussed below.

#### 3.2.1. Characterization of Layer #1 (Ester-Epoxy Resin)

As shown in Appendix A, the ATR-FTIR spectrum of layer #1 matched with that of the epoxy resin mold used for the preparation of the cross-section samples except for an additional carbonyl peak at 1716 cm^−1^. A library search result also produced the best match with ester epoxy resins having ATR-FTIR peaks at 3030, 912, and 826 cm^−1^ due to the υ(C–H), υ(C–O), and δ(C–O–O) bands, respectively, which are characteristic of the oxirane ring of epoxy resins [55,56]. The peaks at 1605, 1581, 1230, 1083, 1008, 937, and 720 cm^−1^ can be attributed to the benzene ring of bisphenol, 1505 and 826 cm^−1^ to p-substituted bisphenol, 1557 and 1540 cm^−1^ to an amino curing agent, 1031 and 1180 cm^−1^ to ether, and 1716 and 1180 cm^−1^ to ester. The X-ray spectrum of this layer (Figure 8a) revealed C, N, O, Al, Si, Ti, Sn, S, and Fe. Al and Si are from a kaolinite mineral based on the ATR-FTIR peaks at 3688, 3623, 1031, 1008, 936, and 912 cm^−1^ matching those of standard kaolinite (Appendix A). The detection of Ti is consistent with the identification of rutile by RMS (peaks at 238, 442, and 609 cm^−1^ in Figure 7). In addition, the Raman peaks at 282 and 638 cm^−1^ probably indicate the presence of FeS_x_ [57] and SnO_2_ [58], respectively, supporting the detection of Sn, Fe, and S by SEM/EDX. The Raman peaks for FeS_x_ and SnO_2_ and the X-ray peaks for Fe, S, and Sn were weak compared to that of TiO_2_, indicating the low contents of FeS_x_ and SnO_2_ in the layer.

#### 3.2.2. Characterization of Layers #2, #3, and #4 (Alkyd-Melamine Resins)

As shown in Appendix A, the ATR-FTIR spectra of layers #2 and #5 of sample A and layers #2, #3, and #4 of sample B, respectively, are similar, which are alkyd-melamine resins, as supported by the ATR-FTIR library spectra with the best match. The characteristic peaks for alkyd and melamine are notated by the arrow and solid heart signs, respectively (Figure 6 and Appendix A) [34,35,39]. The relatively strong ATR-FTIR peaks at 1068 and 982 cm^−1^ in layers #2 and #3 were assigned to a SO_4_^2−^ group from BaSO_4_ (Appendix A), which is also supported by the detection of Ba and S in their X-ray spectra (Figure 8a), even though no Raman signal for BaSO_4_ was observed, probably due to its low Raman sensitivity being overwhelmed by highly sensitive organics (Figure 7). The Al and Si X-ray peaks for layers #2 and #3 suggest the presence of an aluminosilicate mineral, but its speciation based on the ATR-FTIR spectra is not feasible because of its low content (Figure 8a). Small Cu and Cl X-ray peaks for layer #3 suggest the presence of a small amount of CuCl_2_, but it was difficult to detect by RMS (Figure 7) [59]. 

For layer #4, no other elements except for C and O were detected by SEM/EDX (Figure 8a). The overall patterns of the ATR-FTIR spectra of layers #3 and #4 are similar with minor differences in the range of 1087 to 945 cm^−1^ (see the inset (a) in Appendix A). In addition, layer #4 showed a broad peak centered at 750 cm^−1^, whereas layer #3 showed a sharp triplet at 751, 742, and 720 cm^−1^. This is due to the structural differences in the skeletal backbone of the organic molecules in the alkyd-melamine resins (type-I for layer #2 and #3 and type-II for layer #4), which is exhibited more clearly by the Raman spectra (Figure 7).

#### 3.2.3. Characterization of Layer #5 (Butyl Acrylic Resin)

The ATR-FTIR spectrum of layer #5 matched well with those of poly(butyl methacrylate (Appendix A) [47,60,61]. The ATR-FTIR peaks at 1686, 1602, and 700 cm^−1^ suggest υ(aromatic C=O), υ(aromatic C–C=C), and δ(aromatic C-H) out-of-plane bending modes of the aromatic compound, respectively. The Raman spectrum of this layer (Figure 7) is also consistent with the literature spectrum of acrylics [62]. The X-ray spectrum of this layer (Figure 8a) revealed a small Al X-ray signal, most probably due to the Al_2_O_3_ pigment.

### 3.3. Characterization of Sample C

The visible light optical, SE, and ATR-FTIR images of the molded and polished cross-section of sample C (Figure 2c,f, and i, respectively) revealed four layers with different thicknesses. Figure 6, Figure 8b–d, and Figure 9 present representative ATR-FTIR, X-ray, and Raman spectra of these layers, respectively. Based on the spectral data, two kinds of resins, i.e., polybutadiene along with tributyl phosphate (layers #1 and #2) and acrylic-copolymer (layers #3 and #4), were identified as the major organic species in sample C. Talc and CaCO_3_ and rutile were added as the fillers and a pigment, respectively, in layers #1 and #2, and Al metal flakes were present in layer #3. A detailed characterization of each layer is discussed below.

#### 3.3.1. Characterization of Layers #1 and #2 (Polybutadiene Resin)

The ATR-FTIR spectral patterns and peak positions of layers #1 and #2 suggest a mixture of butadiene polymer and tributyl phosphate (TBP), a polymer additive, when compared with the library spectra of polybutadiene and TBP (Appendix A). The FTIR peaks at 1690, 1643, 1454, 1377, 981, and 960 cm^−1^ were assigned to polybutadiene and the peaks at 1377, 1277, 1123, 1070, 1009, and 875 cm^−1^ were attributed to δ(CH_2_), δ(CH_3_), υ(P=O), υ(–C–C–) skeletal, υ(C–O–Ti), υ(C–O), and υ(P–O–C) bands, respectively, which are characteristic of TBP. Their detailed assignment can be found elsewhere [63,64,65]. TBP is used either as a plasticizer in the synthesis of resins, such as polybutadiene, or a pasting agent of pigments. On the other hand, the ATR-FTIR peaks over the ranges of 1454 to 1350 and 1070 to 900 cm^−1^ are much broader than the peaks of polybutadiene at 1445 and 966 cm^−1^. This is because of a broad CO_3_^2−^ peak centered at 1375 cm^−1^ and a Si–O stretching band of talc at 1008 cm^−1^, as shown in the ATR-FTIR spectra of standard CaCO_3_ and talc (Appendix A). In addition, as shown in Figure 9, the Raman spectra confirmed the presence of CaCO_3_ (peaks at 277 and 1084 cm^−1^), talc (192, 357, and 671 cm^−1^), and rutile (237, 440, and 609 cm^−1^). The X-ray spectra of layers #1 and #2 (Figure 8b,c) show C, O, Mg, Al, Si, Ca, Ti, and Fe, where Ca are from CaCO_3_, Mg and Si from talc, and Ti from rutile. The Al intensities are low, which is probably due to an impurity in talc. The X-ray intensities of Mg, Si, Ca, and Ti vary at different locations of layers #1 and #2, indicating heterogeneous, random distributions of granular or elongated CaCO_3_, talc, and TiO_2_ particles, as observed on the SE image in Figure 8b. As shown in Appendix A, the relative intensities of the ATR-FTIR peaks at the ~1000 cm^−1^ (ν(Si-O) band for talc and at the 950 cm^−1^ (out-of-plane ω(C=C–H) band for butadiene vary considerably at different locations, also indicating the heterogeneity of layers #1 and #2.

#### 3.3.2. Characterization of Layers #3 and #4 (Acrylic Resin)

As shown in Appendix A, the ATR-FTIR spectra of layers #3 and #4 match with that of poly(t-butyl methacrylate), except for the additional strong peaks at 1688 and 700 cm^−1^ due to the υ(aromatic C=O) and δ(aromatic C–H) out-of-plane bending modes of aromatic compounds [49]. Layers #3 and #4 are an acrylic copolymer resin containing aromatic rings because a single band at ~1462 cm^−1^ of δ(-CH_3_, >CH_2_) instead of a doublet band at ~1476 and 1458 cm^−1^ was reported to be characteristic of the copolymers [49]. The X-ray spectra shown in Figure 8d indicate the presence of Al and minor Ti and Fe species in layer #3 and no other additives in layer #4. Layer #3 contains Al as Al flakes because of the dramatically enhanced Al X-ray signals and no Raman signal for Al_2_O_3_ was observed on or near the white strips on the SE image in Figure 8b.

## 4. Conclusions

In this study, ATR-FTIR imaging, RMS, and SEM/EDX were applied in combination for a detailed characterization of three samples of car paint chips. The molded and polished cross-sections of the car paint chips maintaining their layered structures were specially prepared for the multi-modal analysis, for which ATR-FTIR imaging, RMS, and SEM/EDX were applied across the cross-sections. Unambiguous molecular speciation of the chemical components within the layers as well as a determination of the physical layered structures were possible using this multi-modal approach. ATR-FTIR imaging can provide information on the molecular species of organic polymer resins and inorganic compounds as well as on their spatial distributions on a micrometer scale. The Raman spectral data, including the far-IR region down to 50 cm^−1^, is a useful complementary tool to ATR-FTIR for the unambiguous identification of organic and inorganic species. Although elemental composition information from SEM/EDX analysis is insufficient for molecular speciation, the detection of chemical elements in the layers is consistent with and supportive of the ATR-FTIR and Raman data for polymer resins, inorganics, and pigments. Five types of polymer resins, such as alkyd, alkyd-melamine, acrylic, epoxy, and polybutadiene resins, were clearly distinguished along with TiO_2_, SnO_2_, FeS_x_, Fe_3_O_4_, CuCl_2_, ZnO, and Al_2_O_3_ as pigments, and kaolinite, talc, pyrophyllite, BaSO_4_, Al_2_(SO_4_)_3_, Zn_3_(PO_4_)_2_, and Al flakes as fillers. Table 1 provides a summary of the chemical species for all the layers in samples A to C. This study provides detailed information on the chemical identities of every layer of three car paint chips. This would be useful for tracing the origin of the car manufacturer when compared to the coating history database of the respective company, which will clearly provide the potential credentials of cars involved in hit-and-run accidents.

Finally, it is worth noting that the heterogeneity within each layer of the car paint chips can be clearly elucidated by this multi-modal approach. Nonetheless, as SEM/EDX, RMS, and ATR-FTIR imaging have different, i.e., submicron, micron, and supermicron, respectively, spatial resolutions, the layer thicknesses and boundaries need to be defined with SEM/EDX analysis. For substantial information on physicochemical properties and the heterogeneity of each layer, the cross-sections of car paint chip samples could be scanned in an automated analysis mode by the three analytical techniques, which would produce a large amount of spectral data for each sample, and thus the application of supervised classification and/or machine learning techniques would be needed for the classification of layers, all of which require further work.

## Figures and Tables

**Figure 1 molecules-24-01381-f001:**
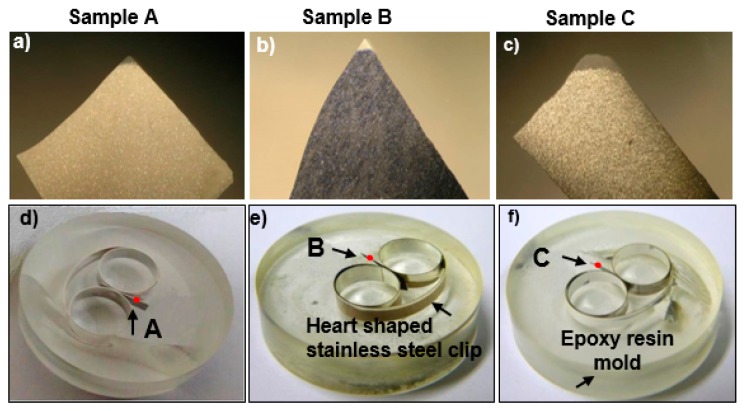
Visible light optical images of (**a**)–(**c**) raw car paint chips and (**d**)–(**f**) molded and polished chips for samples A to C, respectively. Red dots on Figure 1d–f indicate the molded and polished cross sections investigated by ATR-FTIR imaging, RMS, and SEM/EDX.

**Figure 2 molecules-24-01381-f002:**
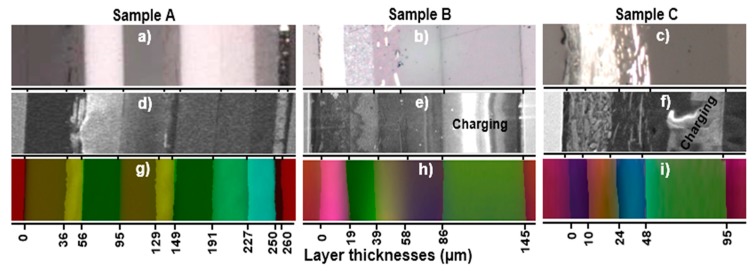
(**a**)–(**c**) Visible light optical images, (**d**)–(**f**) secondary electron images (SEIs) from SEM, and (**g**)–(**i**) ATR-FTIR images of molded and polished cross-sections of samples A to C, respectively. The numbers below the ATR-FTIR images show the layer thicknesses in µm.

**Figure 3 molecules-24-01381-f003:**
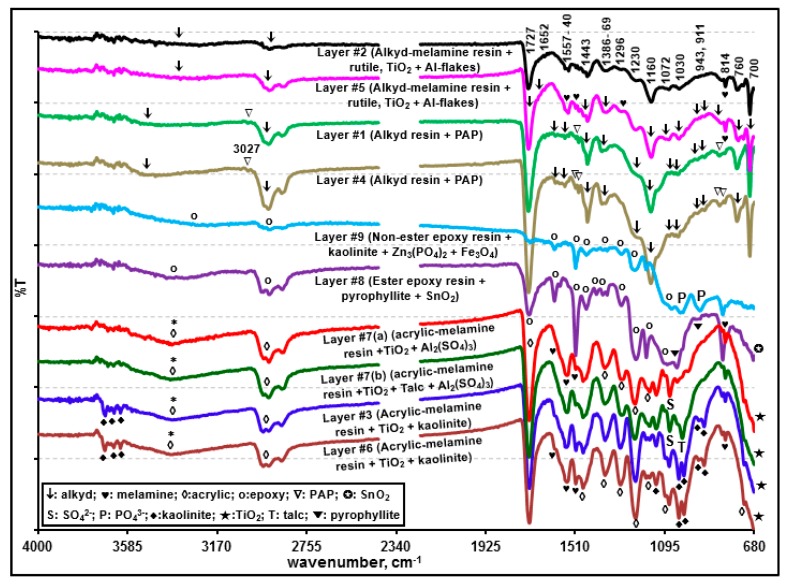
Representative ATR-FTIR spectra of the individual layers of sample A. Peak notations are shown in the inset.

**Figure 4 molecules-24-01381-f004:**
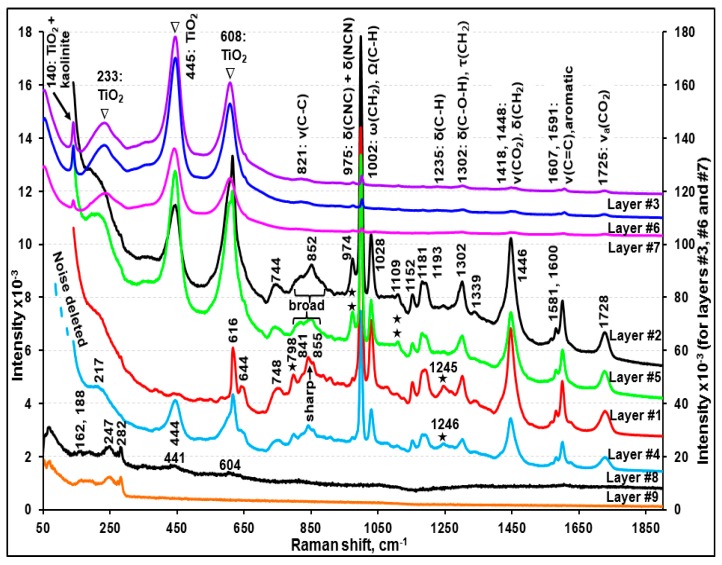
Representative Raman spectra of the individual layers of sample A. The peak notations are v: stretching, a: asymmetric, δ: bending, Ω: out-of-plane, and ε: wagging.

**Figure 5 molecules-24-01381-f005:**
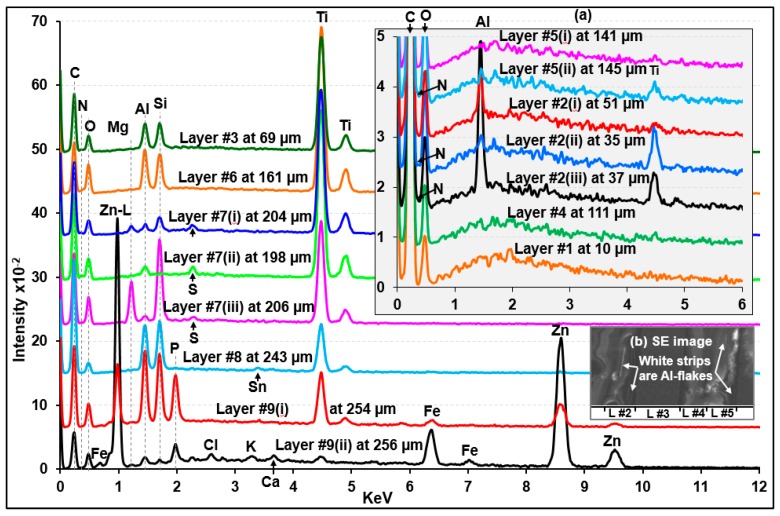
Representative X-ray spectra of the individual layers of sample A.

**Figure 6 molecules-24-01381-f006:**
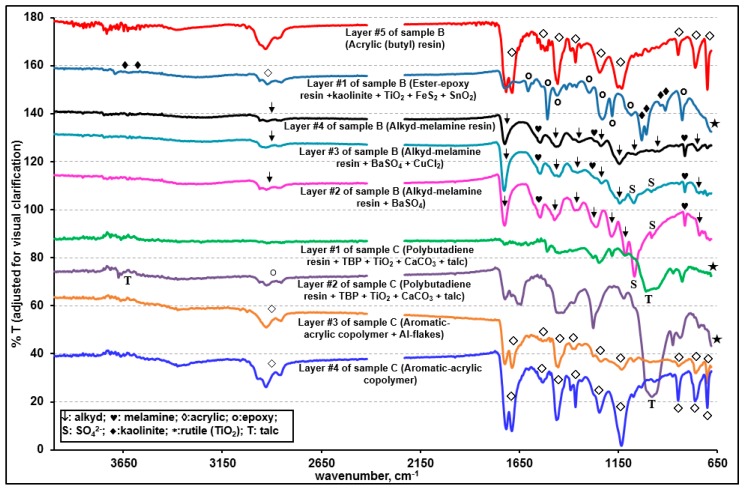
Representative ATR-FTIR spectra of the individual layers of samples B and C. The peak notations are shown in the inset.

**Figure 7 molecules-24-01381-f007:**
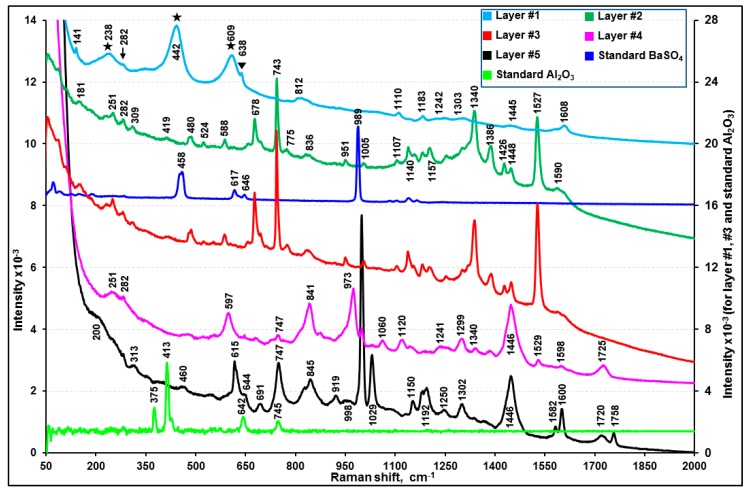
Representative Raman spectra of the individual layers of sample B in comparison with standard BaSO_4_ and Al_2_O_3_.

**Figure 8 molecules-24-01381-f008:**
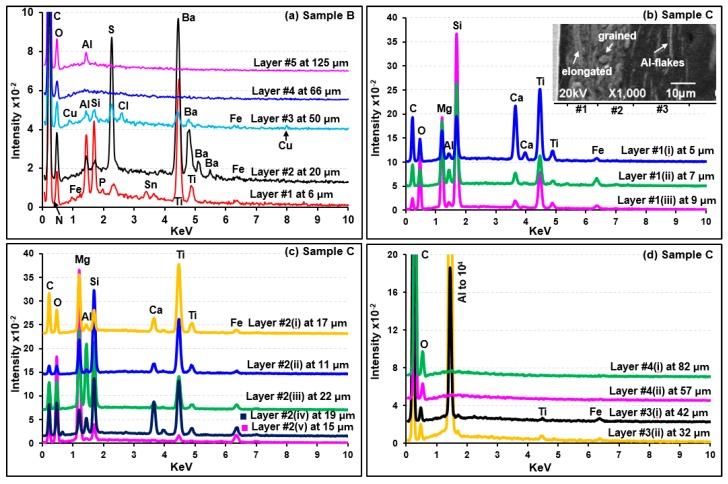
X-ray spectra (**a**) of the individual layers of sample B and (**b**)–(**d**) obtained at various locations in the layers of sample C.

**Figure 9 molecules-24-01381-f009:**
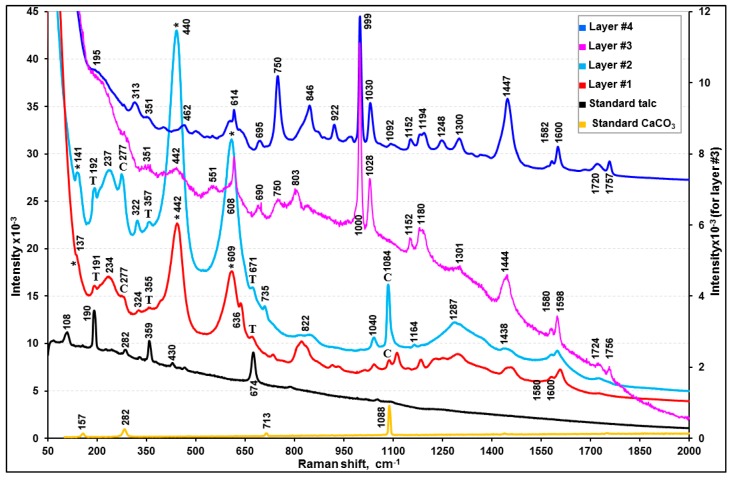
Representative Raman spectra of the individual layers of sample C in comparison with standard talc and CaCO_3_. The peak notations are T: talc; C: CaCO3; *: rutile (TiO_2_).

**Table 1 molecules-24-01381-t001:** Summary of the speciation based on SEM/EDX, ATR-FTIR imaging, and RMS analysis. [TBP: tributyl phosphate].

Sample	Layer #	Elements from EDX	ATR-FTIR and RMS Assignment	
			Major	Minor
**A**	#1	C, O	Enamel or alkyd resin	Polyalkoxylated phenol
	#2	C, N, O, Ti, Al-flakes	Alkyd-melamine resin	Rutile (TiO_2_), Al-flakes
	#3	C, N, O, Ti, Al, Si, Na	Acrylic-melamine resin	Rutile, Kaolinite
	#4	C, O	Enamel or alkyd resin	Polyalkoxylated phenol
	#5	C, N, O, Ti, Al	Alkyd-melamine resin	Rutile, Al-flakes
	#6	C, N, O, Ti, Al, Si, Na	Acrylic-melamine resin	Rutile, Kaolinite
	#7	C, N, O, Ti, Al, Mg, Si, S	Acrylic-melamine resin	Rutile, talc, Al_2_(SO_4_)_3_
	#8	C, N, O, Ti, Al, Si, Zn, Sn	Ester-epoxy resin	Pyrophyllite, SnO_2_
	#9	C, N, O, Al, Si, Ti, Fe, Zn, P	Non-ester epoxy resin	Kaolinite, Zn_3_(PO_4_)_2_
				ZnO, Fe_3_O_4_
**B**	#1	C, N, O, Al, Si, Ti, S, Sn	Ester-epoxy resin	Kaolinite, rutile, SnO_2_
	#2	C, N, O, Ba, S, Al, Si, Fe	Alkyd-melamine resin-I	BaSO_4_
	#3	C, N, O, Ba, S, Cu, Cl	Alkyd-melamine resin-I	BaSO_4_, CuCl_2_
	#4	C, N, O	Alkyd-melamine resin-II	--
	#5	C, N, O, Al	Acrylic (butyl) resin	Al_2_O_3_
**C**	#1	C, O, Mg, Si, Ca, Ti, Al	Polybutadiene + TBP	Talc, CaCO_3_, rutile
	#2	C, O, Mg, Si, Ca, Ti, Al	Polybutadiene + TBP	Talc, CaCO_3_, rutile
	#3	C, O, Al, Ti, Fe	Acrylic copolymer	Al-flakes
	#4	C, N, O	Acrylic copolymer	--

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
