# Peer review of "Multi-Modal Compositional Analysis of Layered Paint Chips of Automobiles by the Combined Application of ATR-FTIR Imaging, Raman Microspectrometry, and SEM/EDX"

_molecules, 2019, doi:10.3390/molecules24071381_

Reviewer 1 Report

This article described the analysis of multi-layered paint chips of hit-and-run cars. ATR-FTIR imaging, Raman microspectrometry RMS, and SEM/EDX techniques were combined for the detailed characterization of car paint chip samples. The authors described that motivation of this manuscript was to combine ATR-FTIR imaging together with SEM/EDX or RMS for the multi-layer car paint analysis.

The works were carried out precisely, but basic methodologies and interpretations were applied for the analyses and the results were not surprising (as expected).

I think the topic may not be interesting for the readers of the special issue in the journal of “Molecules”. Therefore I recommend to submit this article to more specialized journal.

Author Response

Comments (in italics) and responses

This article described the analysis of multi-layered paint chips of hit-and-run cars. ATR-FTIR imaging, Raman microspectrometry RMS, and SEM/EDX techniques were combined for the detailed characterization of car paint chip samples. The authors described that motivation of this manuscript was to combine ATR-FTIR imaging together with SEM/EDX or RMS for the multi-layer car paint analysis.

* Response: We deeply appreciate the time, effort, and interest that the reviewer made/showed.

The works were carried out precisely, but basic methodologies and interpretations were applied for the analyses and the results were not surprising (as expected).

* Response: The sample preparation for the multimodal analysis is completely new. As this multimodal analysis for the multi-layered paint chips has never been done before, it is somewhat surprising to the authors that the reviewer could expect the results from this new multimodal analysis. Even the authors who have worked for years with these three analytical techniques did not expect such detailed results obtainable from these layered samples.

I think the topic may not be interesting for the readers of the special issue in the journal of “Molecules”. Therefore, I recommend to submit this article to more specialized journal.

* Response: In our understanding, this special issue includes articles which have an “analytical chemistry” point of view for the forensic application, therefore, the authors decided to submit our manuscript to this special issue.

Reviewer 2 Report

The authors present a comprehensive compositional analysis of cross sections of layered paint with the end application in forensic analysis of car crashes. The paper is very thorough in terms of the spectroscopic analysis and represents the first application of FTIR imaging, Raman microspectrometry, and scanning electron microscopy/energy dispersive X-ray analysis to this particular application space.  These tools have been shown individually to be useful for paint analysis and together as a multimodal approach to other applications. In general, the paper, as presented is acceptable for publication with a few minor edits and additions.  Specifically,

·        The word color is misspelled to “collar” in the abstract (line 16)

·        The authors talk extensively in the background about ATR-FTIR and SEM/EDX and present a reasonable amount of background information about those techniques, however, RMS is only mentioned in combination with those and not introduced properly.  Please add a few sentences to give the reader a brief background on RMS including some references.

·        As part of the sample preparation, the authors describe a very rigorous polishing protocol. Did the authors study the effect of not polishing the surfaces to such a high degree of smoothness? Is that degree of sample preparation necessary?  Please comment.

·        Figure 3 needs to be larger or split into two figures.  It’s too crowded to really show everything well.

·        Line 267 – contains the word sloppy.  I am uncertain, but I believe this should be “sloping”

·        For this reviewer, it would be helpful if Table 1 were presented at the beginning of the results and discussion then the detailed discussion could refer back to it.

·        For this paper to be impactful, this reviewer would like for the authors to comment on two areas further in the discussion – the effect of heterogeneity and the possibility of automated analysis.  First the authors mention the heterogeneity of the samples, even within a layer and remark that the technique deployed are able to highlight this heterogeneity, particularly in Sample C.  However, sample heterogeneity (especially within a layer) could also complicate layer identification.  There needs to be careful consideration about sampling density for the microprobe methods and spatial resolution for the imaging methods. Could the authors address advantages and or the limitations of this multimodal approach with respect to the ability to characterize heterogeneous samples?  What size inclusions could be seen?  How thin or thick can the layers be?  How well can boundaries be discerned?  Second, presumably for this multimodal approach to be useful in a forensics setting, some type of automated analysis – like supervised classification or machine learning to classify a layer based on a set of spectral features. Have the authors attempted this?  Could they give some idea of whether they believe based on the complexity of the data, this would be feasible?

Author Response

Comments (in italics) and responses

The authors present a comprehensive compositional analysis of cross sections of layered paint with the end application in forensic analysis of car crashes. The paper is very thorough in terms of the spectroscopic analysis and represents the first application of FTIR imaging, Raman microspectrometry, and scanning electron microscopy/energy dispersive X-ray analysis to this particular application space.  These tools have been shown individually to be useful for paint analysis and together as a multimodal approach to other applications. In general, the paper, as presented is acceptable for publication with a few minor edits and additions. 

* Response: We deeply appreciate the time, effort, and interest that the reviewer made/showed.

The word color is misspelled to “collar” in the abstract (line 16)

* Response: The reviewer is right. It was changed and we thank for the correction.

The authors talk extensively in the background about ATR-FTIR and SEM/EDX and present a reasonable amount of background information about those techniques, however, RMS is only mentioned in combination with those and not introduced properly. Please add a few sentences to give the reader a brief background on RMS including some references.

* Response: Indeed, we agree that the general analytical features of RMS as well as of ATR-FTIR imaging and SEM/EDX were not sufficiently described. The following part was added in the revised manuscript (lines: 72-96).

“SEM/EDX can provide information on the physical structures and elemental compositions of micrometer-sized samples with submicron lateral resolution, and yet.it has limited capabilities for performing molecular speciation of particles. Vibrational spectroscopic techniques such as RMS and ATR-FTIR are powerful for functional group analysis and molecular speciation of organic and inorganic chemical compounds including hydrated species under ambient conditions. Although RMS and ATR-FTIR are similar in that they belong to vibrational spectroscopic techniques, their vibrational signals are generated from different fundamentals; i.e. RMS provides information on molecular vibrations based on the difference in wavelength between the incident and scattered visible radiation (Raman scattering), whereas ATR-FTIR based on the attenuation of the evanescent wave generated by the total reflected mid-IR radiation on the IRE crystal. According to selection rules, for IR spectroscopy it is necessary for the molecule to have a permanent electric dipole and for Raman spectroscopy it is the polarizability of the molecule which is important. Therefore, the differences in their spectra owing to their different signal generation mechanisms (i.e. scattering vs absorption of energy) and different selection rules would make two fingerprint techniques rather complementary. RMS and ATR-FTIR imaging provide spectra with a typical spectral range between 100 and 4000 cm-1 and 680-4000 cm-1, respectively, making RMS efficient to identify metal oxides. Further, due to the incident radiation, RMS has better lateral resolution than ATR-FTIR imaging has, so that RMS is more powerful for the investigation of heterogeneity of micrometer-sized samples. The mostly sharp Raman peaks are useful for the unambiguous molecular speciation. On the other hand, laser beam employed in RMS can induce damage on the samples and the interference by the fluorescence often encountered in RMS needs to be minimized, which are not the problem in ATR-FTIR measurements. ATR-FTIR imaging provides ATR-FTIR spectrum at each pixel in image field, which is valuable for obtaining information on the molecular species for the samples, whereas RMS mapping to obtain the spatial, chemical heterogeneity for the samples takes much longer time as RMS images are acquired by point-by-point scanning mode [26-33].”

As part of the sample preparation, the authors describe a very rigorous polishing protocol. Did the authors study the effect of not polishing the surfaces to such a high degree of smoothness? Is that degree of sample preparation necessary?  Please comment.

* Response: As demonstrated in the manuscript, the multi-layered paint chips have micrometer-scale layer structures. In order to unambiguously elucidate those layered structures, it is critical to maintain the exact multi-layer structures so that the multimodal analytical approach can be reliably applied. We did not want to be confused with the heterogeneity which might be induced by the improper sample preparation. We think this sample preparation can ensure the reliable investigation of the multilayer paint chip samples. This argument was added in the revised manuscript (lines: 143-145, “This rigorous sample preparation was performed to maintain exact multi-layer sample structures of micrometer size and thus to ensure the reliable investigation of the samples using the multi-modal approach.” ).

Figure 3 needs to be larger or split into two figures. It’s too crowded to really show everything well.

* Response: Indeed, Fig. 3 looks crowded in the current form. We think that it would look OK when this paper is typeset.  

Line 267 – contains the word sloppy.  I am uncertain, but I believe this should be “sloping”

* Response: Yes, the reviewer is right and “sloppy” was replaced with “sloping”. Thanks again.

For this reviewer, it would be helpful if Table 1 were presented at the beginning of the results and discussion then the detailed discussion could refer back to it.

* Response: We think that it might be a little confusing to put the summary first and describe the way of the identification later. It may be better to put the result summary in the conclusion section.

For this paper to be impactful, this reviewer would like for the authors to comment on two areas further in the discussion – the effect of heterogeneity and the possibility of automated analysis.  First the authors mention the heterogeneity of the samples, even within a layer and remark that the techniques deployed are able to highlight this heterogeneity, particularly in Sample C.  However, sample heterogeneity (especially within a layer) could also complicate layer identification.  There needs to be careful consideration about sampling density for the microprobe methods and spatial resolution for the imaging methods. Could the authors address advantages and or the limitations of this multimodal approach with respect to the ability to characterize heterogeneous samples?  What size inclusions could be seen?  How thin or thick can the layers be?  How well can boundaries be discerned?  Second, presumably for this multimodal approach to be useful in a forensic setting, some type of automated analysis – like supervised classification or machine learning to classify a layer based on a set of spectral features. Have the authors attempted this?  Could they give some idea of whether they believe based on the complexity of the data, this would be feasible?

* Response: As suggested by the reviewer, we added the following arguments at the end of the revised manuscript (lines: 441 -449).

“Finally, it is worth to notice that the heterogeneity within each layer of the car paint chips can be clearly elucidated by this multi-modal approach. Nonetheless, as SEM/EDX, RMS, and ATR-FTIR imaging have different, i.e., submicron, micron, and supermicron spatial resolutions, respectively, the layer thickness and boundaries needs to be defined with SEM/EDX analysis. For substantial information on physicochemical properties and heterogeneity of each layer, the cross-sections of car paint chip samples could be scanned in an automated analysis mode by the three analytical techniques, which would produce a large amount of spectral data for each sample, and thus the application of supervised classification and/or machine learning techniques would be needed for the classification of layers, all of which require further works.”   

Thank the reviewer for the valuable comments again.

Reviewer 3 Report

Molecules - 476572

This paper deals with a multimodal approach to provide confirmatory forensic evidence of car paint chips. The authors provide a thorough discussion and sufficient data to substantiate their findings. This approach has been used in various other contexts and even potentially in this forensic context, but the amount of information provided and systematic analysis will contribute to new knowledge and will be useful for the wider scientific community. The authors may want to address some of the following observations / recommendations / comments:

1.      Various sections in the text refers to “sloppy peaks” e.g. In addition, a sloppy decreasing peak with a peak 267 minimum at 684 cm-1 ascribed by the asterisk indicates TiO2. Sn can be from SnO2 and the trace 268 amount of Zn appears to be from a mineral impurity. Please change that to a better scientific description as it is not clear what the authors mean by sloppy.

2.      Fig 7 needs a legend that will mirror the annotation used in the spectrum.

3.      There are a few typos here and there and the authors are urged to read through the document carefully again.

4.      A link to the colour of the paint chips and the dye identified should be made.

5.      The authors also need to further expand on how the paint chips are different and how this approach will be able to link the evidence to a specific vehicle.

6.      Would they have found differences for the same model of car, from the same manufacturer, but a different year?

When these points have been considered, the paper can be considered for publication in Molecules.

Author Response

Comments (in italics) and responses

This paper deals with a multimodal approach to provide confirmatory forensic evidence of car paint chips. The authors provide a thorough discussion and sufficient data to substantiate their findings. This approach has been used in various other contexts and even potentially in this forensic context, but the amount of information provided and systematic analysis will contribute to new knowledge and will be useful for the wider scientific community. The authors may want to address some of the following observations / recommendations / comments:

* Response: We deeply appreciate the time, effort, and interest that the reviewer made/showed.

1. Various sections in the text refers to “sloppy peaks” e.g. In addition, a sloppy decreasing peak with a peak 267 minimum at 684 cm-1 ascribed by the asterisk indicates TiO2. Sn can be from SnO2 and the trace 268 amount of Zn appears to be from a mineral impurity. Please change that to a better scientific description as it is not clear what the authors mean by sloppy.

* Response: Thank you for the correction. As the reviewer 2 suggested, the “sloppy” word was changed into the “sloping” word.

2.  Fig 7 needs a legend that will mirror the annotation used in the spectrum.

* Response: We think that the annotation used in the spectrum was explained clearly in the text, so that Fig. 7 would be OK as it is.  

3. There are a few typos here and there and the authors are urged to read through the document carefully again.

* Response: We carefully read through the manuscript again. Indeed, we found and corrected typos which were marked in the revised manuscript. Thank the reviewer for the comment.

4. A link to the colour of the paint chips and the dye identified should be made.

* Response: We did not pay much attention to the color of the sample layers as the optical microscopy (OM) could not provide enough color information on each layer. The visible light color of the paint chip layers is only available from the OM, not from ATR-FTIR, RMS, and SEM.

5. The authors also need to further expand on how the paint chips are different and how this approach will be able to link the evidence to a specific vehicle.

* Response: This work is to demonstrate how much detailed information can be obtained from the paint chips using this new multimodal approach in a point of the analytical context in the forensic field. Of course, the next work should expand on how the paint chips are different and how this approach will be able to link the evidence to a specific vehicle as the reviewer points out.

6. Would they have found differences for the same model of car, from the same manufacturer, but a different year?

* Response: Like the previous comment, this comment also can be answered by a further work working with different paint chip samples with different characteristics.

When these points have been considered, the paper can be considered for publication in Molecules.

* Response: We tried to follow the reviewers’ comments and suggestions as much as we can and we appreciate the reviewer’s comments again